# In Vitro Effects on Thrombin of Paris Saponins and In Vivo Hemostatic Activity Evaluation of *Paris fargesii* var. *brevipetala*

**DOI:** 10.3390/molecules24071420

**Published:** 2019-04-11

**Authors:** Feiyan Wen, Tiezhu Chen, Hongxiang Yin, Juan Lin, Hao Zhang

**Affiliations:** 1West China School of Pharmacy, Sichuan University, Chengdu 610041, China; wfyxp963@126.com; 2Sichuan Provincial Key Laboratory of Quality and Innovation Research of Chinese Materia Medica, Sichuan Academy of Chinese Medicine Sciences, Chengdu 610041, China; cctzcd@126.com (T.C.); linjuan_2015@126.com (J.L.); 3College of Ethnomedicine, Chengdu University of Traditional Chinese Medicine, Chengdu 611130, China; hongxiangy@126.com

**Keywords:** Rhizoma Paridis, Paris saponin H, *Paris fargesii* var. *brevipetala*, thrombin, hemostatic activity, UPLC-MS

## Abstract

The resource shortage of Rhizoma Paridis has never been effectively addressed, and the industry continues to search for alternative resources. The in vitro effects on thrombin of Paris saponins and in vivo hemostatic activity of *Paris fargesii* var. *brevipetala* (PF) were evaluated in this study. PF is considered to be an alternative source of Rhizoma Paridis (RP). The in vitro incubation experiment was designed to investigate the effects on thrombin activity of Paris saponin H (PS H) and saponin extract in PF. The bleeding time of mouse tail snipping was used to evaluate the in vivo hemostatic effects of Paris saponins. Also, in vivo changes in four blood coagulation parameters in rats after oral administration of different groups of Paris saponins were compared. The effects of Paris saponins on liver function and blood lipid parameters were examined in order to avoid drug-induced liver injury. Activity studies of thrombin after ultra-filtration centrifugation showed that Paris saponins were able to enhance thrombin activity. Ultra performance liquid chromatography mass spectrometry (UPLC-MS) analysis results of the substrates led us to speculate that there is a specific binding between Paris saponins and thrombin. PS H and Paris saponins in PF significantly shortened the bleeding time in mice. One pathway by which Paris saponins enhance in vivo blood coagulation is by increasing fibrinogen (FIB), among the four blood coagulation parameters in rats. At the same time, the effects on liver and blood lipid parameters were insignificant. *P. fargesii* var. *brevipetala* can be developed as an alternative medicinal source of Rhizoma Paridis.

## 1. Introduction

Rhizoma Paridis (RP) has many important pharmacological activities and is used as an important raw ingredient for many proprietary Chinese medicines. *Paris polyphylla* var. *chinensis* (PPC) and *P. polyphylla* var. *yunnanensis* (PPY) are two legal species [1]. The issue of resource shortage has never been effectively addressed, and the industry continues to search for alternative resources [2,3]. In the early studies, *Paris fargesii* var. *brevipetala* (PF) was found to be widely distributed in western Sichuan, where it is an important folk source of Rhizoma Paridis. The contents of PF saponins, mainly consisting of pennogenin, were found to be high [4], and almost all of them were able to meet the current pharmacopoeia standards. Paris saponin H (PS H, Figure 1a) is a main saponin of pennogenin in PF. Since PF is currently not a legitimate source in the pharmacopoeia, the medicinal value of this variant is yet to be systematically studied and scientifically evaluated in order to facilitate the further development and utilization of this variant. 

The antitumor activity of RP saponins represents one of the hot areas of current research [4,5,6,7,8,9], but RP is mainly used for clinical hemostasis now. For example, RP is the only medicinal material for the hemostatic drug Gongxuening, using pennogenin as the standard substance for quality control [1]. Therefore, hemostasis should be an important index for evaluating the medicinal value of RP. The hemostatic mechanisms of RP saponins have been preliminarily reported [10,11,12,13,14,15,16]. A study by Guo Lin [12] showed that there were significant mutually enhanced effects among pennogenin monomer, dioscin monomer, and two types of steroidal saponins on a uterine smooth muscle contraction model. This may be the chemical and pharmacological basis of Gongxuening using total steroid saponins extracted from RP as the main active ingredient in the treatment of gynecological hemorrhaging. Using animal models, Fu Yali [13] confirmed that the total steroid saponins extracted from RP induced in vivo hemostasis and was able to directly induce platelet aggregation in vitro. The series of results suggested that pennogenins may be a novel class of platelet agonists. 

However, hemostasis and coagulation are a complex process involving many coagulation factors and pathways. Studies conducted by previous project teams preliminarily confirmed the hemostatic activity of RP saponins [16]. In this study, the effects of PF saponin extract and PS H on thrombin activity following in vitro incubation were firstly examined. Secondly, the hemostatic activities of PF saponin extract and PS H were evaluated using a mouse tail snipping technique. Then, the effects of orally administered PF saponin extract and PS H on four blood coagulation parameters were examined to investigate the procoagulant mechanisms in vivo. In addition, the liver is an important organ for material transformation and is where most of the coagulation factors are synthesized. Accordingly, the liver plays a very important role in the processes of hemostasis and coagulation in the body. The literature hasreported that coagulation factors rapidly decreased in the cases of liver injuries including trauma, e.g., liver rupture, leading to coagulopathy. In addition, the liver also synthesizes various kinds of albumins, including fibrinogen; when liver impairment takes place, the level of synthesis declines and the quality deteriorates. Moreover, the antitumor activity and cytotoxicity of RP saponins were obvious, and hepatotoxicity demonstrated a significant dose correlation [17,18]. The study results by Fan Wei [18] confirmed that RP saponins at 350 mg/kg induced severe hepatotoxicity in rats and RP saponins at 50 mg/kg caused varying severity of liver injuries in rats. So, in this study, the effects of PF saponins and PS H at this dose on liver function were also examined while the hemostatic activity was evaluated.

The level of blood lipids plays an important role in the blood coagulation process. Blood lipids and lipoproteins can affect the exogenous coagulation system, endogenous coagulation system, fibrinogen content, and fibrin dissolution, resulting in increased coagulation activity and decreased fibrinolytic activity. More studies showed that an elevated blood lipid level, especially an elevated triglyceride level, was one of the important reasons for changes in coagulation and fibrinolytic activity. Elevated blood lipids can easily lead to diseases such as hypertension and diabetes; therefore, it is necessary to control blood lipid parameters while enhancing coagulation. While exploring the mechanisms of Paris saponin-related coagulation, the effects of PS H at this dose on blood lipid parameters were also examined in this study.

Overall, the hemostatic activity of *P. fargesii* var. *brevipetala* was evaluated to expand the medicinal sources of Rhizoma Paridis. An in vitro study on thrombin activity aimed to explore the hemostatic mechanism of Paris saponins. The effects of Paris saponins on liver function and blood lipid parameters were examined in order to avoid drug-induced liver injury.

## 2. Results and Discussion

### 2.1. Thrombin In Vitro Tests

Thrombin consists of two peptide chains (31 kD and 6 kD) through a disulfide bond, and has a molecular weight of 37 kD. During the coagulation process, thrombin catalyzes the hydrolysis of fibrinogen; then, it cleaves peptides A and B in the fibrinogen to become fibrin monomer with significantly reduced solubility. These molecules can spontaneously associate with each other to form insoluble fibrin, which is the last step of coagulation. Activation of thrombin by the drug enhanced the coagulation process to achieve hemostasis.

By observation, we found coagulation did not occur in the inactivated thrombin + PS H group until half an hour had passed, while the other three groups coagulated within 1 min. The clotting time in the other three groups is shown in Figure 1b. It can be clearly seen that RP saponins showed a trend of enhancing the agglutination of thrombin and fibrinogen. With higher saponin content in the PF saponin extract, the tendency became more obvious in group M+Q (thrombin + PF saponin extract). The PF saponin extract contained PS H, PS VI, and PS VII.The result of group M+Q suggests that pennogenin, other than PS H, also had similar enhancing effects on the agglutination of thrombin and fibrinogen. 

A UPLC-MS precursor ion was used to scan PS H, and the determined results of the substrate after incubation and ultra-filtration in each group are as shown in Figure 1c. Chromatogram A is the control article of PS H, chromatogram B is the inactivated thrombin + PS H group, and chromatogram C is the thrombin + PS H group. PS H was significantly lower in group C than in group B, suggesting that thrombin specifically binds to PS H after the portion of non-specific binding of inactivated thrombin with PS H has been subtracted. The results further verified the effects of PS H on thrombin activity. It is speculated that after PS H specifically binds to thrombin, it enhances the agglutination between thrombin and fibrinogen, further promoting the occurrence of blood coagulation; this means that it has similar effects as a thrombin agonist. It is speculated that one of procoagulant pathways for RP saponins is to promote the agglutination of thrombin and fibrinogen through the activation of thrombin activity, thereby enhancing coagulation. 

### 2.2. Hemostasis Effects in Mice

The bleeding time and hemostatic effect in each group are shown in Figure 2. When the treatment group and positive group were compared with the blank group, the bleeding time was significantly shortened. The differences had statistical significance (*p* < 0.05) and there was a certain dose correlation. When treatment groups were compared with the positive group, the bleeding time presented no obvious difference (*p* > 0.05). The hemostatic effects in the PS H high-, medium-, and low-dose groups and in the PF extract high-dose group were greater than in the Yunnan Baiyao positive group. The in vivo study with mice confirmed the significant ability of PS H and the plant extract to stop bleeding.The effect in group Q32 (32 mg/kg PF saponin extract) was significantly higher than that in other groups, suggesting that other RP saponins (PS VII and PS VI, etc.) in this pennogenin-dominated extract also had hemostatic effects. This is contrary to the results for hemostatic activity shown in reference [10], where some *Paris* species plants except *P. fargesii* var. *fargesii* were found to significantly shorten the tail bleeding time and blood clotting time. 

PS H was reported [8] to be widely contained in various *Paris* species, such as *Paris yunnanensis*, *Paris fargesii*, *Paris mairei*, *Paris thibetica*, *Paris axialis*, *Paris bashanensis*, and *Paris verticillata*. In the authors’ earlier study, with high content, PS H is found to be a main saponin of pennogenin in PF. It provided enough amount of PS H monomer for this activity research after being separated and purified. Fu [19] identified pennogenin glycosides as the active ingredients of *P. polyphylla* Sm. var. *yunnanensis* in promoting hemostasis in vivo. Chen [20] also reported that pennogenin glycosides were responsible for the hemostatic effect of *Trillium kamtschaticum*. Contents of Paris saponins in *P. fargesii* and *Trillium tschonoskii* were reportedly rather higher than that in the other *Paris* species [21]. *Trillium* species were used as a fake version of Rhizoma Paridis because of the lack of resources and the extremely high price. Even the stems and leaves of *Paris* were developed as the new medicinal parts to address resource scarcity [22,23]. In this study, PF extract, mainly containing pennogenin saponins, demonstrated a significant hemostatic effect, which provided an activity study basis for PF to serve as a medicinal Rhizoma Paridis resource.

### 2.3. Effects on Four Items of Blood Coagulation, Liver Function, and Blood Lipid Parameters in Rats

After intragastric administration of PS H and PF extracts every day for five days, the effects of on the four items of blood coagulation in rats of all groups are presented. It can be seen in Figure 3 that PS H and PF extracts had no significant effects (*p* > 0.05) on shortening the normal prothrombin time (PT), activated partial thromboplastin time (APTT), and thrombin time (TT) in rats. In some cases, the values were slightly higher than those in the blank group. This is different from the previous result that RP saponins reduced the PT and APTT but had no significant effect on TT [10]. However, groups with RP saponins and the positive group significantly increased fibrinogen (FIB) (*p* < 0.05, compared with blank control). The main component of PF saponin extract was found to be pennogenin in the authors’ earlier study. It is speculated that pennogenin exerts its hemostatic activity by increasing FIB as one pathway of the hemostatic mechanism. 

After intragastric administration of PS H and PF extracts every day for 10 days, the liver function and blood lipid parameters in rats of all groups are presented. It was found that RP saponins only slightly increased liver function and blood lipid parameters, but they remained within the normal range (Figure 4). Each treatment group and the positive group were compared with the blank group, and the parameters presented no obvious difference (*p* > 0.05). This suggests that RP saponins under this dose have minimal effects on liver function and blood lipid parameters during the hemostasis process. The study results by Fan Wei [18] confirmed that RP saponins at 350 mg/kg induced severe hepatotoxicity in rats and RP saponins at 50 mg/kg caused varying severity of liver injuries in rats. In this study, the effects of PF saponins and PS H at 20 mg/kg for 10 days on liver function were examined, which manifested as a low risk of drug-induced liver injury. Inrecent years, many steroidal saponins have been isolated from the rhizomes of PPY, and their side effects of uterine contractile activity are light and few too [24].

## 3. Materials and Methods 

### 3.1. Materials 

Kunming mice and Wistar rats, half male and half female, were purchased from Jianyang Dashuo animal Science and Technology Corporation (Jianyang, China). The rhizome of PF was collected from Pengzhou, Sichuan, China and identified by Professor Zhang as *P. fargesii* var. *brevipetala*. The specimen was kept in the specimen room of West China School of Pharmacy, Sichuan University (Sichuan, China). PS H was extracted and separated from the rhizome of PF by the authors at the early stage and was identified according to the physical and chemical properties and spectral data with a purity of over 98%. The preparation method of PF extract is shown below, and the content of total saponins was more than 65% by HPLC.The characteristics of the equipment and protocol used for quantification by HPLC were the same as those in reference [4]. The rhizome was crushed and extracted by hot leaching (45 °C) with 70% ethanol (15 times the volume). Then, the concentrated extract was dispersed and dissolved with water to be separated by a HPD100 macroporous resin column, eluted with water, 30% ethanol, 50% ethanol, 70% ethanol, and 95% ethanol, successively. The crude extracts of the 30% ethanol part mainly contained polysaccharides and mucus, the 50% ethanol part mainly contained Paris saponins of higher polarity, the 70% ethanol part mainly contained pennogenin saponins, and the 95% ethanol part was mainly diosgenin saponins. The crude extracts of the 70% ethanol part with the highest total weight were concentrated to be used in this study.

Yunnan Baiyao capsules (ZBA1404) and Gongxuening capsules (ZAA1405) produced by Yunnan Baiyao Co., Ltd. (Yunnan, China) were purchased. All were suspended with 0.2% CMC-Na solution for later use. Ultrapure water (Millipore, MA, USA) were used; thrombin (bovine, MPBIO, Cat. No: 0219990710); fibrinogen (bovine serum, coagulation content 51%, National Institutions for Food and Drug Control, 140626-201310); Na_2_HPO_4_ (chemically pure); and Roche biochemical kit batches ALT 34940601, AST 0764957, ALB 33963501, CHOL 37119001, TG35653001, TP 34777001, HDL24974701, and LDL 31313101were also used. 

### 3.2. Instruments

A blood coagulation analyzer (SYSMAX ca-530/ca-7000, Kobe, Honshu, Japan); automatic biochemical analyzer (Hitachi 7100/7180, Tokyo, Honshu, Japan); low-temperature centrifuge (Thermo, Waltham, MA, USA); sodium citrate vacuum blood vessel collection equipment (2 mL, batch no. 201406005, Shanghai Aoxiang medical group, Shanghai, China); and 0.5 mL 10 kD ultra-filtration centrifuge tube (Millipore, MA, USA)were used in the experiments.

The chromatographic conditions were as follows: Waters Acquity UPLC/Quattro Premier XE mass spectrometer (Waters, Milford, MA, USA); Acquity UPLC^TM^BEH C18 column (1.7 mm, 2.1 mm × 100 mm, Waters); the mobile phase was composed of water (*v*/*v*) (10%) and methanol (90%) isocratic elution with aflow rate of 0.25 mL/min. Electrospray ionization mass spectrometry (ESI-MS) analyses were performed on a Waters Quattro Premier XE mass spectrometer equipped with an ESI source. The optimized ESI source parameters were as follows: Capillary voltage, 2.8 kV (positive mode); extractor voltage, 5 V; cone voltage, 40 V; source temperature, 100 °C; desolvation temperature, 250 °C; flowrate of the desolvation gas (N_2_), 600 L/h; and cone gas, 40 L/h. The mass spectra were recorded using the single ion reaction (SIR) mode on a mass of *m*/*z* = 893.5 in positive ion mode. All the operations, acquisition, and data analyses were controlled using MassLynx V4.1 software (Waters, Milford, MA, USA).

### 3.3. Methods

#### 3.3.1. Methods of Investigating RP Saponins’Effects on Thrombin Activity.

Thrombin and RP saponins were incubated in vitro. After centrifugation and ultra-filtration, thrombin, unbound PS H, and salt impurities were separated. Then, a fibrinogen method was used to determine and compare thrombin activity after incubation; the filtrate was analyzed with UPLC-MS and the inactivated thrombin group was compared with the RP saponins group to subtract non-specific binding in order to determine whether there was specific binding between PS H and thrombin. This method referred to the determination of a thrombin titer in Chinese Pharmacopoeia, and the actual enzyme titer (unit) was logarithmically correlated with the clotting time (s). 

Thrombin (90–300 U/mg, 4 mg) was obtained and precisely weighed to prepare 5 mL thrombin solution (72–240 U/mL) with 0.9% sodium chloride solution in order to control the clotting time properly within 14–60 s, and the solutions were freshly prepared prior to use. Preparation of the fibrinogen solution was undertaken as follows: ~98 mg fibrinogen (coagulum content 51%) was obtained and precisely weighed to prepare 25 mL fibrinogen solution containing 0.2% coagulum using 0.9% sodium chloride solution; after the pH was adjusted to 7.0–7.4 with 0.05 mol/L disodium hydrogen phosphate solution, it was diluted to a 50 mL solution containing 0.1% coagulum using 0.9% sodium chloride. The solutions were freshly prepared prior to use. A quantity of 5 mg PS H was obtained and precisely weighed to prepare 50 mL of 100 μg/mL stock solution using ultrapure water; 1 mL stock solution was precisely measured to be diluted to 10 mL in order to prepare 10 μg/mL of PS H solution for later use. In order to prepare PF extract, after being finely ground, 10 mg fine powder was obtained and precisely weighed to prepare 100 mL of 100 μg/mL stock solution with ultrapure water; 2 mL stock solution was precisely measured and diluted to 10 mL in order to prepare 20 μg/mL extract solution, and this was kept below 4 °C for later use. 

The experiment was divided into four groups, namely, thrombin + PS H, inactivated thrombin + PS H, thrombin + extract, thrombin + ultrapure water, and three samples were tested in parallel in each group. A quantity of 100 μL of thrombin solution was measured, then 100 μL each of the above four solutions was added; after incubating in a 37 °C water bath for 2 h, the mixture was quantitatively transferred into an ultra-filtration centrifuge tube and centrifuged to about 100 μL, 3500 r/min, ambient conditions. After washing twice with 100 μL ultrapure water, it was centrifuged again until the enzyme solution was concentrated to about 100 μL; the enzyme solution was quantitatively collected and diluted to 1 mL for the comparison of thrombin activity. The filtrate was quantitatively transferred and diluted to 1 mL for the UPLC-MS detection; PS H was determined using a precursor ion scanning technique. 

A 0.9 mL quantity of fibrinogen solution was added to each of the tubes with an inner diameter of 1 cm and length of 10 cm; after incubating in a 37 ± 0.5 °C water bath for 5 min, 0.1 mL of each of the above four thrombin solutions was precisely measured and quickly added to each of the above test tubes. Timing started immediately; the test tubes were shaken evenly and placed in a 37 ± 0.5 °C water bath to observe the initial clotting time of fibrin, and each concentration was measured at least five times to calculate the mean (the difference between the maximum and minimum of the five measurements should not exceed 10% of the mean; otherwise, the measurement should be repeated). The clotting time of each solution was compared. 

#### 3.3.2. Bleeding Time (BT) in Mice (Tail Snipping Method) 

Fifty-six Kunming mice (20–25 g), half males and half females, were randomized into seven groups (*n* = 8). The grouping and dosing were as follows: Blank control group; positive control group (B, Yunnan Baiyao 1 g/kg); PS H high-dose group (H32, 32 mg/kg), PS H medium-dose group (H20, 20 mg/kg),and PS H low-dose group (H8, 8 mg/kg); and PF extract high-dose group (Q32, 32 mg/kg) and low-dose group (Q16, 16 mg/kg). Intragastric administration was applied to all groups for three consecutive days, and an equal volume of 0.2% CMC-Na solution was given to the blank control group. Two hours after the last dose, the mice were immobilized and mouse tails were severed at 3 mm from the tip with a pair of sharp scissors. Timing started when bleeding occurred. The blood was absorbed with filter paper once every 30 s until the bleeding stopped spontaneously. We stopped timing when no blood was visible on the filter paper, and this time was the bleeding time (BT). The experiments were carried out following the ethics approval number: SCXK (Chuan) 5015-030 approved by the bioethics committee.

Hemostatic effect = 100% × (blank group BT − drug group BT)/blank group BT(1)

#### 3.3.3. Four Items of Blood Coagulation and Blood Lipid Parameters

Sixty-four Wistar rats, half males and half females, weighing 110–130 g, were randomized into eight groups with eight rats in each group. For the PS H high- (H20, 20 mg/kg), medium- (H12.5, 12.5 mg/kg), and low- (H5, 5 mg/kg) dose groups; the PF extract high- (Q20, 20 mg/kg) and low- (Q10, 10 mg/kg) dose groups; the two positive control groups (B, 0.6 g/kg; G, 80 mg/kg); and the blank control group(C, 0.2% CMC-Na), the samples were all suspended with 0.2% CMC-Na. The compound was intragastrically administered every day for five days. Two hours after the last dose, the animals were anesthetized to collect 2 mL blood from the abdominal aorta. The vacuum blood collection tubes containing sodium citrate anticoagulant were centrifuged at 3500 r/min for 10 min at room temperature, and the plasma was obtained for the determination of the coagulation constants PT, TT, APTT, and FIB. The PT, APTT, and TT were determined by the solidification method, and FIB was determined by the Cluss method using a blood coagulation analyzer. The reagents and quality control products were imported matching products.

Another 64 Wistar rats, half males and half females, weighing 80–120 g, were randomized into eight groups, eight rats in each group. The dosing and grouping were the same as before. The compound was intragastrically administered every day for 10 days; two hours after the last dose, the animals were anesthetized to collect 2 mL blood from the abdominal aorta. The serum was diluted for the determination of liver function and blood lipid parameters, including ALT, AST, ALB, CHOL, TG, TP, HDL, and LDL. An automatic biochemical analyzer was applied to measure the above parameters by using the related biochemical kits (Roche).

### 3.4. Data Processing

Statistical analysis of the data (available in Appendix A) was conducted using SPSS statistics 17.0 data processing software (SPSS, Chicago, IL, USA). For normally distributed parameters, independent sample *t*-testing was used for inter-group comparison; differences were considered to have statistical significance when *p* < 0.05. The data were plotted with Excel not only to calculate the effects of RP saponins on the bleeding time after the tails were severed but also to determine their impact on the four items of blood coagulation, liver function, and blood lipid parameters in rats. 

## 4. Conclusions

The pharmacological activity of *P. fargesii* var. *brevipetala* (PF) is mainly related to the chemical constituents of saponins, and the results of previous studies by the authors [4] showed high contents of total saponins contained in PF, mainly pennogenin. Paris saponin H, VI, and VII contained in PF, are widely contained in the genus *Paris* and have presented extensive activities, especially in antitumor, cytotoxicity, and hemostasis activity [4,25]. The hemostatic effects of Paris saponin H and PF saponin extract were examined in this study. In vitro experiments were firstly developed and applied to study the effects of PF saponins on thrombin activity, which confirmed that PF saponins were able to enhance thrombin activity. As a local hemostatic drug, thrombin is not easily obtained, and Paris saponins may be developed into an auxiliary medication. In vivo experiments also demonstrated the significant hemostatic activity of PF saponins in mice and presented their apparent ability to stop bleeding. Determination of the four items of blood coagulation showed that increasing fibrinogen was a pathway in their procoagulant mechanism. In this study, only minimal effects on liver function and blood lipid parameters were found, and alow risk of drug-induced liver injury manifested.Therefore, *P. fargesii* var. *brevipetala* can be developed as an alternative medicinal source of Rhizoma Paridis.

## Figures and Tables

**Figure 1 molecules-24-01420-f001:**
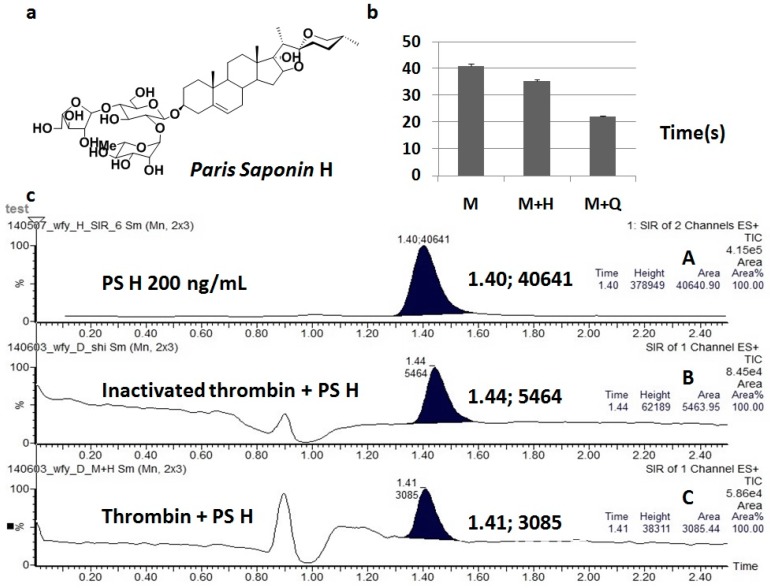
Effect of Paris saponin H (PS H) and *Paris fargesii* var. *brevipetala* (PF) saponin extract on thrombin; (**a**) structural formula of PS H; (**b**) coagulation time in vitro (*n* = 5); M: Thrombin + ultrapure water; M+H: Thrombin + PS H; M+Q: Thrombin + PF saponin extract; coagulation did not occur in the inactivated thrombin + PS H group; (**c**) UPLC-MS assay results of the substrate from each group after incubation and ultra-filtration; chromatogram A is the control article of PS H (200 ng/mL); chromatogram B is the inactivated thrombin + PS H group; and chromatogram C is the thrombin + PS H group.

**Figure 2 molecules-24-01420-f002:**
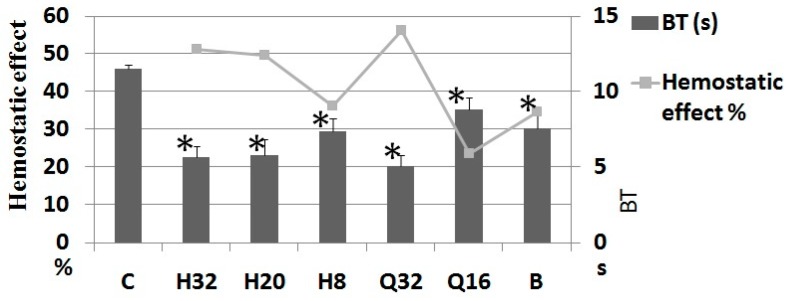
Bleeding time (BT) in mice and hemostasis effect of PS H and PF saponin extract (*n* = 8); hemostatic effect = 100% × (blank group BT − drug group BT)/blank group BT; C: Control (0.2% CMC-Na solution); H: PS H (H32, 32 mg/kg; H20, 20 mg/kg; H8, 8 mg/kg); Q: PF saponin extract (Q32, 32 mg/kg; Q16, 16 mg/kg); B: Yunnan Baiyao (1 g/kg); * compared with 0.2% CMC-Na (*t*-test *p* < 0.05).

**Figure 3 molecules-24-01420-f003:**
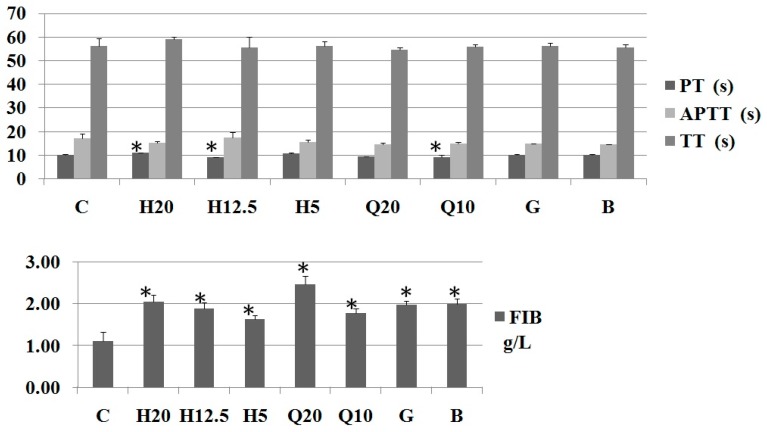
Effects on the four blood coagulation parameters of PS H and PF saponin extract in rats (*n* = 8); C: Control (0.2% CMC-Na solution); H: PS H (H20, 20 mg/kg; H12.5, 12.5 mg/kg; H5, 5 mg/kg); Q: PF saponins extract (Q20, 20 mg/kg; Q10, 10 mg/kg); G: Gongxuenin (80 mg/kg); B: Yunnan Baiyao (0.6 g/kg); * compared with 0.2% CMC-Na (*t*-test *p* < 0.05); prothrombin time (PT); thrombin time (TT); activated partial thromboplastin time (APTT); fibrinogen (FIB).

**Figure 4 molecules-24-01420-f004:**
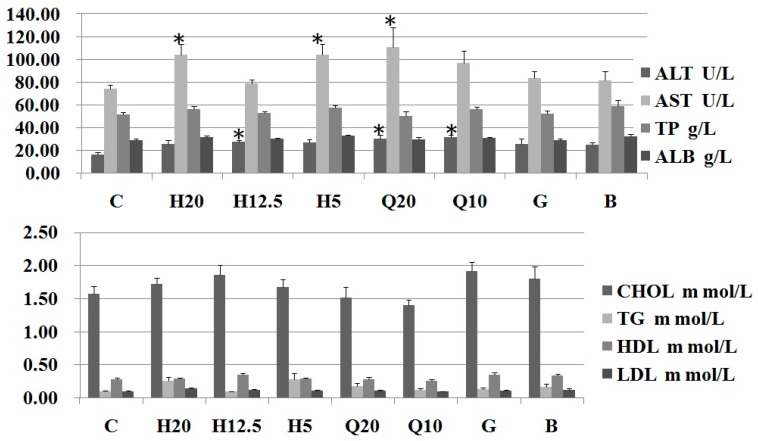
Effects on the liver function and blood lipid parameters in rats (*n* = 8) of PS H and PF saponin extract; C: Control (0.2% CMC-Na solution); H: PS H (H20, 20 mg/kg; H12.5, 12.5 mg/kg; H5, 5 mg/kg); Q: PF saponins extract; G: Gongxuenin (80 mg/kg); B: Yunnan Baiyao (0.6 g/kg); * compared with 0.2% CMC-Na (*t*-test *p* < 0.05); alanine aminotransferase (ALT), aspartate aminotransferase (AST), albumin (ALB), total cholesterol (CHOL), triglyceride (TG), total proteins (TP), high-density lipoprotein (HDL), and low-density lipoprotein (LDL).

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
