# Peer review of "In Vitro Effects on Thrombin of Paris Saponins and In Vivo Hemostatic Activity Evaluation of Paris fargesii var. brevipetala"

_molecules, 2019, doi:10.3390/molecules24071420_

Round 1

Reviewer 1 Report

The authors evaluated the "Effects on Thrombin in vitro of Paris Saponins and Hemostatic Activity Evaluation in vivo of Paris fargesii var. brevipetala". They did a great work from an experimental point of view.

However, the presentation of results should be improved as the text does not flow well. Some sentences are too long and difficult to understand: For example:

"After being intragastrically administered every day for 5 days, the results can be seen in

Figure 3 that PS H and PF extracts had no significant effects on shortening normal prothrombin".

I suggest the authors sending the MS to an English native speaker.

Moreover, discussion section should be improved by adding some references supporting the findings obtained. In general, the number of references should be increased.

Conclusions are vague and should be expanded. Moreover, please, avoid abbreviatures in this section.

Author Response

Response to Reviewer 1 Comments

Point 1: However, the presentation of results should be improved as the text does not flow well. Some sentences are too long and difficult to understand: For example:

"After being intragastrically administered every day for 5 days, the results can be seen in Figure 3 that PS H and PF extracts had no significant effects on shortening normal prothrombin".

I suggest the authors sending the MS to an English native speaker.

Response 1: We have our manuscript checked by a professional English editing service. For example: " After intragastric administration of PS H and PF extracts every day for 5 days, the effects of on the four items of blood coagulation in rats of all groups are presented. It can be seen in Figure 3 that PS H and PF extracts had no significant effects(p>0.05) on shortening the normal prothrombin time (PT)

Point 2: Moreover, discussion section should be improved by adding some references supporting the findings obtained. In general, the number of references should be increased.

Conclusions are vague and should be expanded. Moreover, please, avoid abbreviatures in this section.

Response 2: Discussion section in the revised file has been improved by adding references, so the number of references has been increased. Conclusions have be expanded and abbreviatures are changed in this section.

Reviewer 2 Report

However there are some information that should be clarified in the manuscript (major revisions) in order to be accepted for publication.

The aims of the study should be reformulated according to the research objectives and target results. The conclusions must reflect the innovation of this study and the perspectives.

The text may be handled by a native English speaker.

I didn’t lack new information on other industrial properties of carotenoids (eg. antimicrobial properties). Please describe.

What is the industrial use of results?

The statistics are unclear. Which tests were used (figure 3 and 4)

Materials and methods section is incomplete and tables containing results should be improved (Headings and subtitles are unclear. State all instrumentation and chemicals/reagents used in a single subtitle. Describe analytical steps clearly for both techniques).

A very poorly described method

Bleeding time (BT) in mice (tail snapping method)

Was the analytical method validated (UPLC-MS)? Give information?

(repeatability, reproducibility, linearity, regression comparison)

Author Response

Response to Reviewer 2 Comments

Point 1: The aims of the study should be reformulated according to the research objectives and target results. The conclusions must reflect the innovation of this study and the perspectives.

Response 1: The aims of the study and the conclusions have been revised in the file.

Point 2: The text may be handled by a native English speaker.

Response 2: We have our manuscript checked by a professional English editing service.

Point 3: I didnt lack new information on other industrial properties of carotenoids (eg. antimicrobial properties). Please describe. What is the industrial use of results?

Response 3: Because of its significant activity, utilization of the Rhizoma Paridis is expanding sharply which leads to the wild source decreasing rapidly. Paris is perennial plant and can only be harvested after 5-7 growth, which aggravates the shortage of Paris source. Furthermore, affected by SARS in the year of 2003, the price of Rhizoma Paridis has kept rising from RMB 120-140 Yuan /kg to 1000-1400 Yuan /kg. The resource shortage of Rhizoma Paridis has never been effectively addressed, and the industry continues to search for alternative resources. Quality evaluation of Paris fargesii var. brevipetala has been researched in our earlier study. Now hemostatic activity of Paris fargesii var. brevipetala was evaluated to expand the medicinal sources of Rhizoma Paridis.

Study on thrombin activity in vitro aims to explore the hemostatic mechanism of Paris saponins. As a local hemostatic drug, thrombin was not easy to be obtainedParis saponins may be developed to an auxiliary medication. The effects of Paris Saponins on liver function and blood lipid parameters were examined in order to avoid drug-induced liver injury.

Point 4: The statistics are unclear. Which tests were used (figure 3 and 4)

Response 4:   3.4 Data processing independent sample t-test was used for inter-group comparison” The statistics tests were added in figures.

Point 5: Materials and methods section is incomplete and tables containing results should be improved (Headings and subtitles are unclear. State all instrumentation and chemicals/reagents used in a single subtitle. Describe analytical steps clearly for both techniques).

Response 5: Tables containing results were available in part of supplementary materials. Materials and methods section has been completed. Headings and subtitles are clearly stated. More analytical steps for techniques were described in the file.

Point 6: A very poorly described method           Bleeding time (BT) in mice (tail snapping method)

Response 6: The method was redescribed as: Fifty-six Kunming mice (20–25g), half males and half females, were randomized into seven groups (n=8). The grouping and dosing were as follows: blank control group; positive control group (B, Yunnan Baoyao 1 g/kg); PS H high-dose group (H32, 32 mg/kg), PS H medium-dose group (H20, 20 mg/kg), and PS H low dose group (H8, 8 mg/kg);and PF extract high-dose group (Q32, 32 mg/kg) and low-dose group (Q16, 16 mg/kg). Intragastric administration was applied to all groups for 3 consecutive days, and an equal volume of 0.2% CMC-Na solution was given to the blank control group. Two hours after the last dose, the mice were immobilized and mouse tails were severed at 3 mm from the tip with a pair of sharp scissors. Timing started when bleeding occurred. The blood was absorbed with filter paper once every 30 sec until the bleeding stopped spontaneously. We stopped timing when no blood was visible on the filter paper, and this time was the bleeding time (BT).The experiments were carried out following the ethics approval number: SCXK ()5015-030 approved by the bioethics committee.

Point 7: Was the analytical method validated (UPLC-MS)? Give information?    (repeatability, reproducibility, linearity, regression comparison)

Response 7: Preliminary conclusions are obtained by comparing peak areas of groups. RSD (relative standard deviation) of peak areas in parallel samples are relatively high. The analytical method is not verified in this study. Quantitative methods including analytical method validate and different concentration levels of PS H experimental are planning to design and improve experimental methods in the next study. 

Reviewer 3 Report

Molecules 475984

Effects on Thrombin in vitro of Paris Saponins and Hemostatic Activity Evaluation in vivo of Paris fargesii var. brevipetala

The work should be reviewed in some of its sections and then it could be considered for acceptance

Some suggestions are given below:

1-      The aims of the manuscript (lines 57-62; 71-72 page 1)  should be specified at the end of the introduction.

2-      The paragraph Lines 90-95 should be revised

As shown by the results of this study, coagulation did not occur in inactivated thrombin + PS H group. (Where?).

Please indicate where this result is shown

3-      Figure 1c

Please include A, B and C in the respective chromatograms

4-       3.2.2 Bleeding time (BT) in mice (tail snapping method)

A paragraph should be included with relevant information regarding in vivo tests should be included

“The experiments were carried out following  Protocol number approved by bioethics committee or relevant institution”

5-      The conclusions should be reviewed.

The following paragraph refers to previous investigations of the authors, of others or of this work?

“the results of previous studies show a high saponin content in PF,

 mainly pennogenin”.

6-      3.1 Materials and Instruments

Lines 120-121 page 5 :”The preparation method PF extract is shown as  below and the content of total saponins was more than 65% by HPLC.”

The characteristics of the equipment and protocol used for quantification by HPLC should be mentioned.

7-      3.1 Materials and Instruments

Lines 122- 125 page 5: “Then the concentrated extract was dispersed and dissolved with water to be separated by the macroporous resin column of HPD100, eluted with water, 30% ethanol, 50% ethanol, 70% ethanol and 95% ethanol successively.

The crude extracts of the 70% ethanol parts was concentrated to used in this study.”

The justification for the selection of this extract should be mentioned

Author Response

Response to Reviewer 3 Comments

Point 1: The aims of the manuscript (lines 57-62; 71-72 page 1) should be specified at the end of the introduction.

Response 1: The aims of the manuscript have been specified at the end of the introduction in the revised file.

Point 2: The paragraph Lines 90-95 should be revised

As shown by the results of this study, coagulation did not occur in inactivated thrombin + PS H group. (Where?). Please indicate where this result is shown.

Response 2: By observing, coagulation did not occur in the inactivated thrombin + PS H group until half an hour had passed, while the other three groups coagulated within 1 min. Stop observing. The paragraph has been revised in the file.

Point 3: Figure 1c       Please include A, B and C in the respective chromatograms

Response 3:    Markers include A, B and C in the respective chromatograms were adjusted to a distinct position in Figure 1c.

Point 4: 3.2.2 Bleeding time (BT) in mice (tail snapping method)

A paragraph should be included with relevant information regarding in vivo tests should be included

The experiments were carried out following Protocol number approved by bioethics committee or relevant institution”

Response 4: The paragraph has been included in vivo tests.

The experiments were carried out following the ethic approval number: SCXK () 5015-030 approved by bioethics committee”

Point 5: The conclusions should be reviewed.

The following paragraph refers to previous investigations of the authors, of others or of this work?

the results of previous studies of the authors show a high saponin content in PF, mainly pennogenin”.

Response 5: The conclusions have been reviewed. The following paragraph refers to previous investigations of the authors earlier work. the results of previous studies by the authors [4] showed high contents of total saponins contained in PF, mainly pennogenin.”

Point 6: 3.1 Materials and Instruments

Lines 120-121 page 5:”The preparation method PF extract is shown as below and the content of total saponins was more than 65% by HPLC.”

The characteristics of the equipment and protocol used for quantification by HPLC should be mentioned.

Response 6: The characteristics of the equipment and protocol used for quantification by HPLC were the same in reference [4]. The specific description is as followsHPLC analysis on Ps were carried out on an Agilent 1200 liquid chromatograph system (Agilent Technologies, USA), equipped with a quaternary pump, an online degasser, and a column temperature controller, coupled with an ELSD (Alltech 3300Alltech Associates, USA) as the detector. The column temperature was kept at 30. The samples were separated with a Kromasil RP-C18 column (4.6 mm×250 mm, 5 μm, Sweden) using water (A) and acetonitrile (B) under gradient conditions (0–5 min, linear gradient 30–45% B; 5–10 min, isocratic 45% B; 10–20 min, linear gradient 45–50% B; 20–30 min, linear gradient 50–60% B; 30–35 min, linear gradient 60–100% B; each run was followed by a 5 min wash with 100% B and an equilibration period of 10 min with 30% B; ) as the mobile phase at a flow rate of 1 mL/min. The drift tube temperature for ELSD was set at 45 , and the nebulizing gas flow rate was 1.6 L/min.

Ten standards and all samples were filtered through 0.45 μm filter (Anpel Co., Shanghai, China) before injected into HPLC. The injection volume was 20 μL. Every sample solution was injected in triplicate, and the contents of the samples were determined from the corresponding calibration curves.

Point 7: 3.1 Materials and Instruments

Lines 122- 125 page 5: “Then the concentrated extract was dispersed and dissolved with water to be separated by the macroporous resin column of HPD100, eluted with water, 30% ethanol, 50% ethanol, 70% ethanol and 95% ethanol successively. The crude extracts of the 70% ethanol parts was concentrated to used in this study.”

The justification for the selection of this extract should be mentioned.

Response 7: The justification for the selection of this extract has been mentioned in the revised file. Skeleton type of Paris saponins is various. The content of total saponins in PF is high and the solubility of total extract is poor. After segmentation by macroporous resin, 30% ethanol part mainly contains polysaccharides and mucus, 50% ethanol part mainly contains Paris saponins of higher polarity, 70% ethanol part mainly contains pennogenin saponins, and 95% ethanol part is mainly diosgenin saponins. The crude extracts of the 70% ethanol part with highest total weight was concentrated to be used in this study.

Round 2

Reviewer 1 Report

The authors addressed well my comments. Therefore, I consider the MS is acceptable for publication in Molecules in its present form.